# Deleterious Interaction between the Neurosteroid (3α,5α)3-Hydroxypregnan-20-One (3α,5α-THP) and the Mu-Opioid System Activation during Forced Swim Stress in Rats

**DOI:** 10.3390/biom13081205

**Published:** 2023-08-01

**Authors:** Giorgia Boero, Minna H. McFarland, Ryan E. Tyler, Todd K. O’Buckley, Samantha L. Chéry, Donita L. Robinson, Joyce Besheer, A. Leslie Morrow

**Affiliations:** 1Bowles Center for Alcohol Studies, School of Medicine, University of North Carolina at Chapel Hill, 3027 Thurston Bowles Bldg., CB 7178, Chapel Hill, NC 27599, USA; giorgia_boero@med.unc.edu (G.B.);; 2Department of Psychiatry, School of Medicine, University of North Carolina at Chapel Hill, Chapel Hill, NC 27599, USA; 3Department of Pharmacology, School of Medicine, University of North Carolina at Chapel Hill, Chapel Hill, NC 27599, USA

**Keywords:** neurosteroids, behavior, stress, forced swim stress, 3α,5α-THP, allopregnanolone, opioid system, mu-opioid receptor, CTAP, brexanolone

## Abstract

The neurosteroid 3α,5α-THP is a potent GABA_A_ receptor-positive modulator and its regulatory action on the HPA axis stress response has been reported in numerous preclinical and clinical studies. We previously demonstrated that 3α,5α-THP down-regulation of HPA axis activity during stress is sex-, brain region- and stressor-dependent. In this study, we observed a deleterious submersion behavior in response to 3α,5α-THP (15 mg/kg) during forced swim stress (FSS) that led us to investigate how 3α,5α-THP might affect behavioral coping strategies engaged in by the animal. Given the well-established involvement of the opioid system in HPA axis activation and its interaction with GABAergic neurosteroids, we explored the synergic effects of 3α,5α-THP/opiate system activation in this behavior. Serum β-endorphin (β-EP) was elevated by FSS and enhanced by 3α,5α-THP + FSS. Hypothalamic Mu-opiate receptors (MOP) were increased in female rats by 3α,5α-THP + FSS. Pretreatment with the MOP antagonist D-Phe-Cys-Tyr-D-Trp-Arg-Thr-Pen-Thr-NH2 (CTAP; 2 mg/kg, IP) reversed submersion behavior in males. Moreover, in both males and females, CTAP pretreatment decreased immobility episodes while increasing immobility duration but did not alter swimming duration. This interaction between 3α,5α-THP and the opioid system in the context of FSS might be important in the development of treatment for neuropsychiatric disorders involving HPA axis activation.

## 1. Introduction

Eighty years after the neurosteroid was identified in the adrenal gland [1], the FDA approved (3α,5α)3-hydroxy-pregnan-20-one (3α,5α-THP, also known as allopregnanolone or its commercial formulation, brexanolone) for the treatment of post-partum depression (PPD) [2]. According to the *Diagnostic and Statistical Manual of Mental Disorders* (DSM–5), PPD is deemed present when a patient has a major depressive episode along with the onset of pregnancy or within four weeks of delivery and five depressive symptoms are present for at least two weeks [3]. The FDA’s decision highlights the therapeutic importance of 3α,5α-THP signaling in the brain, suggesting the use of 3α,5α-THP treatment in other neuropsychiatric disorders, such as major depression, general anxiety disorder, post-traumatic stress disorder and various substance use disorders. All these diseases share a dysregulation of the balance between the glutamatergic excitatory inputs and GABAergic inhibitory transmission across the brain [4]. 3α,5α-THP is a potent positive modulator of GABA_A_ receptors, and its ability to attenuate stress-induced effects has been well reported in the past [5,6,7]. Moreover, several stress-related diseases show an increase in toll-like receptor-mediated inflammatory neuroimmune signaling. Recent studies from our lab demonstrated that 3α,5α-THP inhibits pro-inflammatory signals in rat brains, a mouse macrophage cell line, human macrophages in cell culture and human blood samples from PPD patients [8,9,10,11].

We previously demonstrated that 3α,5α-THP regulation of the hypothalamic-pituitary-adrenal (HPA) axis is both sex- and region-dependent [12]. 3α,5α-THP modulates the HPA axis stress response at many different levels, depending on sex and type of stressor [13]. In this study, we investigated the effects of 3α,5α-THP on rat behavior during forced swim stress (FSS). The 3α,5α-THP dose we used in our experiments (15 mg/kg) has known anxiolytic and anti-convulsant actions [14,15] but no hypnotic effect in rats or mice [16,17]. The data revealed a surprising outcome of 3α,5α-THP treatment combined with FSS—an increase in submersion and immobility and a decrease in swimming behavior. 

Previous studies have demonstrated that activation of the HPA axis leads to the synthesis and release of endogenous opioid peptides [18], including β-endorphin (β-EP) [19]. In fact, in response to a stress stimulus, the paraventricular nucleus of the hypothalamus produces and releases corticotropin-releasing factor (CRF), the main activator of the HPA axis, in response to stress. From the hypothalamus, CRF travels to the anterior pituitary where it stimulates the transcription of the pro-opiomelanocortin (POMC) gene, precursor to adrenocorticotrophic hormone (ACTH) and β-EP [20]. Released into the bloodstream, ACTH reaches the adrenal glands, where it induces the synthesis and release of corticosteroids, while β-EP reaches the Mu-opioid receptors (MOP) to exert its functions. It has been shown that hypothalamic CRF neurons receive direct inhibitory inputs from arcuate nucleus β-EP neurons [21]. In addition, β-EP indirectly inhibits CRF activity by inhibiting norepinephrine neurons that stimulate CRF neurons [22]. Some studies suggest that endogenous opioid activation in response to stress might be a protective mechanism to avoid stress-induced impairments [23,24,25]. Likewise, it has been shown that forced swim stress induces an increase in β-endorphin in female rats [26]. Moreover, it has been previously demonstrated that 3α,5α-THP modulates central opioid expression and produces opioid inhibition over HPA responses [27]. Therefore, we hypothesized that the behavioral outcome observed in FSS might be the result of the modulation of opioid-mediated inhibitory transmission by 3α,5α-THP. 

In the present study, we recorded the behavior of male and female rats over 10 min of FSS after administering 15 mg/kg of 3α,5α-THP, followed by molecular analysis of opioid system components, such as circulating β-endorphins and hypothalamic Mu-opioid receptors (MOP). To establish whether 3α,5α-THP had a sedative effect at this dose, we assessed locomotor activity in a separate group of animals. Finally, we scored the behavior during the FSS, pre-treating the animals with the MOPs antagonist D-Phe-Cys-Tyr-D-Trp-Arg-Thr-Pen-Thr-NH2 (CTAP) at two different doses (0.5 and 2 mg/kg, respectively), 30 min before administering 3α,5α-THP. Our results suggest that 3α,5α-THP + FSS activates the Mu-opioid receptor system to a different extent in male and female rats, but overall, 3α,5α-THP interaction with the opioid system affected the animals’ behavior during the FSS. This could be a clinically relevant discovery in relation to the therapeutic applicability of 3α,5α-THP in neuropsychiatric disorders that involve concurrent opioid system activation. 

## 2. Materials and Methods

### 2.1. Animals

All animal use was approved by the Institutional Animal Care and Use Committee at the University of North Carolina at Chapel Hill. Male and female (PN80-100) Sprague–Dawley rats used in all experiments were bred in-house from Envigo stock (Envigo, Indianapolis, IN, USA). A total of 90 male and 94 female rats were used to perform the experiments in this study. To avoid fluctuations in endogenous 3α,5α-THP due to the estrus cycle, and given that the main progesterone peak in the rat estrous cycle occurs in the early evening of proestrus and returns to basal levels by the morning of estrus [28,29], all experiments were conducted between 8 a.m. and 12 p.m. Rats were pair-housed and kept in a temperature- and humidity-controlled vivarium on a 12-h light/dark cycle with free access to water and food. All animals were handled for 10 to 14 days prior to the start of the experiment to minimize handling and injection stress (Figure 1). 

### 2.2. Drug Administration

#### 2.2.1. 3α,5α-THP

3α,5α-THP was prepared as previously described in Boero et al., 2022 [13]. Briefly, hydroxypropyl-β-cyclodextrin (45% *w*/*v* in water) was prepared by dissolving 3α,5α-THP at a concentration of 7.5 mg/mL and administered to the rats via intraperitoneal (IP) injection. Solutions were prepared the day before the experiment and kept under stirring at 4 °C overnight. Animals were randomly divided into two groups, and according to their weight, all rats received an IP injection of 3α,5α-THP (15 mg/kg) (Steraloids Inc, Newport, RI, USA, #P38000-000) or an equivalent volume of vehicle (VEH). This dose was chosen based on our previous work [13] and is known to have an anxiolytic and anti-convulsant effect [14,15] but no hypnotic effect [16,17]. Fifteen minutes after VEH or 3α,5α-THP administration, the rats were subjected to forced swim stress for ten minutes. The rats were euthanized via decapitation 45 min after 3α,5α-THP administration; brains and blood were collected and were immediately frozen at −80 °C until the assay. Data from the VEH groups are described as the baseline measure in experiments.

#### 2.2.2. CTAP 

D-Phe-Cys-Tyr-D-Trp-Arg-Thr-Pen-Thr-NH2 (CTAP) was used as a selective antagonist of Mu-opioid receptors (MOPs) in experiment 3. CTAP (Tocris Bioscience, Bristol, UK, #1560) was dissolved in 1 mL of sterile MilliQ water, following the instructions from the manufacturer. The animals were randomly divided into two groups, and according to their weight, all rats received an IP injection of CTAP (0.5 or 2 mg/kg) or an equivalent volume of vehicle (water, w). The lower dose was chosen based on previous data [30], while the higher dose was chosen based on our preliminary data. It has been demonstrated that CTAP blocks morphine’s maximal effect at the lowest dose (0.5 mg/kg) after 45 min [30]. In our experiments, 30 min after CTAP or water administration, all animals received a second IP injection of 3α,5α-THP or vehicle (see 3α,5α-THP section) and 15 min later, each rat was subjected to the behavioral test. In this way, the animals started the behavioral test at the CTAP maximum effect, 45 min later.

### 2.3. Behavioral Tests

#### 2.3.1. Forced Swim Stress (FSS)

Sprague–Dawley rats were subjected to FSS (experiment 1: males n = 40 and females n = 42; experiment 3: males n = 42 and females n = 44, 80 to 100 days old; see Animal section). The animals swam in a standard clear plastic cylinder (45.7 cm H × 20 cm in diameter; Stoelting Co., Wood Dale, IL, USA, #60160), filled with water to a depth ~30 cm, so that the animals could neither touch the bottom with their tails nor escape from the top. Between each test, the cylinders were thoroughly cleaned. The water temperature was kept at 25 ± 1 °C. Animals were randomly assigned to control or FSS groups. Fifteen min after 3α,5α-THP or vehicle administration, rats were gently picked up by the tail from the home cage and rapidly placed into the middle of the cylinder. The test lasted for 10 min, then the animals were dried with absorbent towels and returned to their home cage for about 20 min. All animals were first-time swimmers and none were used for multiple FSS exposures. To prevent any adverse incidents, animals were closely monitored, and any rat that showed distress for more than 2 sec was removed from the cylinder immediately. One animal was not able to complete the 10 min task and was excluded from the data analysis. Rats were euthanized via decapitation 45 min after 3α,5α-THP or vehicle administration (20 min after the end of the FSS exposure). The entire FSS session was videotaped for subsequent behavioral analysis.

#### 2.3.2. Locomotor Activity

Rats (experiment 2: males n = 8 and females n = 8) were handled for 3 to 5 days prior to the start of the experiment to minimize handling stress. All rats received 2 days of vehicle injections to habituate them to the holding cage, and injection. 3α,5α-THP doses (vehicle, 7.5 mg/kg, and 15 mg/kg) were delivered in a counterbalanced order with a 48-h wash out period between injections. During wash-out periods, animals were left undisturbed in their home cages. On the day of the experiment, rats were injected with vehicle or 3α,5α-THP IP and, 15 min later, gently placed in a chamber (floor space: 42 cm × 42 cm). Locomotor activity was performed for 10 min with lights on and the white noise of the room ventilation, at room temperature. The chambers were equipped with cameras to record locomotor activity for subsequent behavioral analysis. 

### 2.4. Behavioral Analysis

The 10 min FSS sessions were videotaped and subsequently analyzed. Immobility and swimming behavior were evaluated using ANY-maze video tracking system 7.20 (Stoelting Co., Wood Dale, IL, USA). Submersion, climbing, and exploration behavior were analyzed blindly by a trained observer. Duration (sec) and total number of episodes for submersion, immobility, swimming, climbing, and exploration behavior were calculated. We also compared the rats’ position inside the cylinder, measuring the angle (α) during immobility 5 min after the beginning of the FSS for each rat.

(1)Definition of submersion. Submersion was scored when the entire body of the animal, including the head, was upright but underwater, with no possibility of the rat breathing. If the rat showed distress underwater for longer than 2 s, it was removed from the cylinder immediately. Submersion was considered an involuntary passive behavior.(2)Definition of immobility. A rat was considered immobile when floating passively, keeping its head above the surface, making only the necessary movements to maintain balance. For the analysis, immobility behavior was measured starting from the first bout of immobility lasting longer than 1 s. Immobility was considered a passive behavior.(3)Definition of swimming. Swimming was identified as active swimming movements, more than were necessary to maintain the head above the water. Swimming was considered an active behavior.(4)Definition of climbing. Climbing was defined as the animal making upward-directed movements of the forepaws against the walls of the cylinders, trying to escape. Climbing was considered an active behavior.(5)Definition of exploration. Exploration was identified if the animal intentionally swam underwater, exploring the environment looking for a way to escape. Exploration was considered an active behavior.(6)Angle during immobility. The angle (α) during immobility was defined as the position of the animal with respect to the water surface, as previously described by Chen et al., 2015 [31]. One frame was taken out from each video 5 min after the beginning of the test, at the first moment the rat stopped swimming and was in full profile view. The angle (α) was calculated by drawing a triangle between the water surface, the base of the rat tail and the intersection of the rat body with the water using Snip & Sketch tool on Microsoft Windows 10 Enterprise.

### 2.5. Biomarker Measurement

#### 2.5.1. Sample Collection

Twenty min after the end of the FSS exposure, rats from experiments 1 and 3 (males n = 40; females n = 42) were euthanized via decapitation, and blood and whole brains were collected. The whole brain was extracted and frozen at −80 °C until the immunoblotting or ELISA assays were performed. The blood samples were collected in glass tubes (BD Vacutainer Serum, Becton, Dickinson and Company, Franklin Lakes, NJ, USA, #366430), allowed to clot on ice, then centrifuged at 1750× *g* for 15 min at 4 °C. Serum was transferred to Eppendorf Flex 2 mL tubes and frozen at −80 °C until the ELISA assay. 

#### 2.5.2. Immunoblotting

Whole brains were dissected and prepared as previously described for immunoblotting or ELISA [13]. The hypothalamus was homogenized and sonicated in ice-cold CelLytic MT lysis buffer (Sigma-Aldrich, St. Louis, MO, USA, #C3228) with 1× HALT protease and phosphatase inhibitor (Thermo Fisher Scientific, Waltham, MA, USA, #1861281). The lysates were subsequently left on ice for 30 min and then centrifugated at 14,000× *g* for 30 min at 4 °C. Lysate supernatants were collected and transferred to new tubes, and the total protein concentrations were determined via bicinchoninic acid assay (BCA, Thermo Fisher Scientific #23228, #1859078). The proteins (40 μg/lane) were denatured at 95°C for 5 min in LDS sample buffer (2.5 µL per 10 µL sample) (Thermo Fisher Scientific #NP0007) and with sample reducing agent (1 µL per 10 µL sample) (Thermo Fisher Scientific #NP0009) and were separated via NuPAGE™ 10% Bis-Tris Midi Protein Gel (Thermo Fisher Scientific #WG1202, #WG1203) electrophoresis at 125V for 10 min and 165V for the remainder of the process, then transferred to polyvinylidene fluoride membranes (iBlot2 PVDF regular stacks, Thermo Fisher Scientific #IB24001) via the iBlot 2 Dry Blotting System (Thermo Fisher Scientific). Membranes were blocked with 5% Blotting-Grade Blocker (Bio-Rad, Hercules, CA, USA, #1706404) in PBS-T (0.5% Tween-20) for 2 h (room temperature) and subsequently incubated overnight at 4 °C with an anti-MOP antibody (1:1000, R&D System, Minneapolis, MN, USA, #MAB6866). The blots were washed 3 times (15 min each, room temperature) in PBS-T (0.5%) after incubation with primary antibody and subsequently incubated with horseradish peroxidase-labeled secondary anti-mouse IgG antibody (1:2000, Cell Signaling Technologies, Danvers, MA, USA, #7076s) for 1h at room temperature. After another wash in PBS-T (0.5%), the immunoreactive bands were visualized with Western Lightning Plus (Perkin Elmer, Waltham, MA, USA, #NEL105001EA) and detected with enhanced chemiluminescence (ImageQuant LAS4000, GE Healthcare, Amersham, UK). Bands were analyzed using the software program ImageQuant TL v8.1.0.0. The ratio of each protein was calculated by dividing the densitometric measurement of the protein of interest by the corresponding β-Actin. All immunoblotting results are presented as % change vs vehicle no stress ± SEM. 

#### 2.5.3. ELISA

Serum β-endorphin (β-EP) (MyBiosource Inc, San Diego, CA, USA, #MBS763627) was measured via enzyme-linked immunosorbent assay (ELISA) following the manufacturer’s instructions. Results are expressed in nanograms/milliliter (ng/mL). The minimum detectable quantity is 15.625 pg/mL. 

### 2.6. Statistical Analysis

Statistical analysis was performed using GraphPad Prism 9 (GraphPad Software, Inc., Boston, MA, USA). Multiple groups of data were analyzed using a one-way or two-way analysis of variance (ANOVA) and significant interactions were followed up with Tukey’s Honest Significant Differences (HSD) test. A value of *p* < 0.05 was considered statistically significant.

## 3. Results

### 3.1. Experiment 1: FSS Behavioral Analysis 

#### 3.1.1. 3α,5α-THP Induced Submersion Behavior in Male and Female Rats Exposed to FSS

In both male and female animals, 3α,5α-THP administration increased the number of episodes of submersion behavior (Treatment F (1, 36) = 50.81, *p* < 0.0001; Sex F (1, 36) = 1.849, *p* = 0.1824; Interaction between factor F (1, 36) = 1.962, *p* = 0.1698) (Figure 2a).

#### 3.1.2. 3α,5α-THP Also Increased the Duration of Immobility in Male and Female Rats during the Swim Stress

We calculated the duration and the number of episodes that the rats were passively immobile, floating inside the cylinder. There was no difference in the number of times the animals showed immobility behavior between male and female rats (Treatment F (1, 36) = 3.778, *p* = 0.0598; Sex F (1, 36) = 0.4651, *p* = 0.4996; Interaction between factors F (1, 36) = 0.8794, *p* = 0.3546). However, the total time in immobility was higher in the 3α,5α-THP-treated groups (Treatment F (1, 35) = 31.09, *p* < 0.0001; Sex F (1, 35) = 0.02754, *p* = 0.8692; Interaction between factors F (1, 35) = 2.287, *p* = 0.1395) (Figure 2b). 

#### 3.1.3. 3α,5α-THP Affected the Position of the Body in Water during Immobility

During the behavioral analysis, we observed remarkable differences in the animals’ floating body posture. We quantified these differences by calculating the angle between the surface of the water and the animals’ tail–nose axis during the first immobile episode 5 min after the beginning of the FSS (see materials and methods 4.a.6). 

In both male and female rats, α was significantly narrowed in the rats treated with 3α,5α-THP (Treatment F (1, 36) = 21.78, *p* < 0.0001; Sex F (1, 36) = 1.160, *p* = 0.2887; Interaction between factors F (1, 36) = 0.06033, *p* = 0.8074). These differences in body posture were positively correlated with submersion behavior, indicating the increase in α predicted the increase the number of episodes in male (Linear Regression α degrees/number of episodes: R^2^ = 0.6251, F(1, 8) = 13.34, *p* = 0.0065; α degrees/total time: R^2^ = 0.4566, F(1, 8) = 6.72, *p* = 0.0320) and female rats (Linear Regression α degrees/number of episodes: R^2^ = 0.7501, F(1, 8) = 24.02, *p* = 0.0012; α degrees/total time: R^2^ = 0.499, F(1, 8) = 7.969; *p* = 0.0224) (Figure 3).

This result suggests that the change in the position of the body in the water induced by 3α,5α-THP contributes to the increase in submersion behavior. 

#### 3.1.4. 3α,5α-THP Decreased the Total Swimming Time in Male and Females Rats during FSS

The total time spent swimming was measured as a parameter of active behavior. Consistent with the increase in immobility time, 3α,5α-THP administration induced a decrease in swimming time in both male and female rats (Treatment F (1, 36) = 32.38, *p* < 0.0001; Sex F (1, 36) = 0.04369, *p* = 0.8356; Interaction between factors F (1, 36) = 2.535, *p* = 0.1201) (Figure 4a).

#### 3.1.5. 3α,5α-THP Did Not Affect Climbing in Male and Female Rats during FSS

Climbing was measured as another parameter of active behavior. 3α,5α-THP administration did not alter the number of climbing episodes (Treatment F (1, 36) = 2.772, *p* = 0.1046; Sex F (1, 36) = 0.1432, *p* = 0.7074; Interaction between factors F (1, 36) = 0.09163, *p* = 0.7639) or the total time for climbing (Treatment F (1, 36) = 0.9446, *p* = 0.3376; Sex F (1, 36) = 0.2973, *p* = 0.5889; Interaction between factors F (1, 36) = 1.240, *p* = 0.2729) (Figure 4b).

#### 3.1.6. 3α,5α-THP Reduced Exploring Behavior in Female, but Not in Male Rats

In male rats, the 3α,5α-THP did not significantly change exploring behavior. In contrast, in female animals, 3α,5α-THP administration dramatically decreased the number of exploration episodes (Treatment F (1, 36) = 15.03, *p* = 0.0004; Sex F (1, 36) = 0.1701, *p* = 0.6825; Interaction between factors F (1, 36) = 1.531, *p* = 0.2240) and duration (Treatment F (1, 36) = 11.03, *p* = 0.0021; Sex F (1, 36) = 1.708, *p* = 0.1995; Interaction between factors F (1, 36) = 3.636, *p* = 0.0645) (Figure 4c).

### 3.2. Experiment 2: Locomotor Activity Analysis

#### 3α,5α-THP Did Not Affect Locomotor Activity in Male and Female Rats

It has been demonstrated that a 15 mg/kg dose of 3α,5α-THP has anxiolytic and anti-convulsant effects [14,15] but no hypnotic effect [16] in rats. To determine whether the behavioral effects observed during FSS could be attributed to sedative actions of 3α,5α-THP on locomotor activity, we recorded distance traveled after IP administration of 2 different 3α,5α-THP doses (7.5 mg/kg and 15 mg/kg) in male and female rats. The results showed that, in both male and female rats, the distance traveled was not influenced by either dose of 3α,5α-THP (Treatment F (2, 42) = 2.123, *p* = 0.1324; Sex F (1, 42) = 3.919, *p* = 0.0543; Interaction between factors F (2, 42) = 0.08655, *p* = 0.9173). This result suggests that 3α,5α-THP alone did not affect locomotor activity in male and female rats (Figure 5).

### 3.3. Experiment 1: FSS Biochemical and Molecular Analysis

#### 3.3.1. 3α,5α-THP Exacerbated the Increase in β-Endorphins Induced by FSS

FSS behavioral analysis showed that 3α,5α-THP induced an impairment in the animals’ ability to swim, increasing the passive behaviors of submersion and immobility. We hypothesized that the combination of 3α,5α-THP administration and stress may have activated another inhibitory circuit, such as the opioid system. To evaluate this, we measured β-endorphin (β-EP) serum levels after 3α,5α-THP administration in male and female rats exposed to forced swim stress.

In both male and female rats, the exposure to FSS significantly raised β-EP serum levels. In the male group, the animals treated with 3α,5α-THP and exposed to swim stress 15 min later showed a greater increase in β-EP serum levels (Treatment F (1, 27) = 5.246, *p* = 0.0300; Stress F (1, 27) = 88.78, *p* < 0.0001; Interaction between factors F (1, 27) = 10.99, *p* = 0.0026). Similar effects were observed in the female group, where FSS induced an increase in β-EP serum levels and the combination of stress with 3α,5α-THP administration resulted in a greater effect on β-EP serum concentrations in female rats (Treatment F (1, 29) = 12.72, *p* = 0.0013; Stress F (1, 29) = 105.3, *p* < 0.0001; Interaction between factors F (1, 29) = 11.56, *p* = 0.0020). 

Interestingly, β-EP serum levels showed a higher baseline in male vs. female rats (445.2 pg/mL vs. 200.8 pg/mL, *p* < 0.01). Moreover, the stress effect was greater in male than female rats, probably due to the higher baseline (Males FSS VEH = 1093 pg/mL vs. Females FSS VEH = 366.8 pg/mL, *p* < 0.0001; Males FSS 3α,5α-THP = 1688 pg/mL vs. Females FSS 3α,5α-THP = 535.2 pg/mL, *p* < 0.0001) (Figure 6a).

#### 3.3.2. Changes in β-EP Levels Are Induced Specifically by Swim Stress

As previously demonstrated in female rats [26], the results showed that forced swim stress increased circulating β-EP levels. Interestingly, the increase in β-EP serum levels observed after the FSS was not detected following restraint stress. In fact, in both male and female animals, restraint stress did not induce any changes in β-EP. Moreover, 3α,5α-THP administration per se did not affect β-EP concentration (Males Treatment F (1, 27) = 0.05877, *p* = 0.8103; Stress F (1, 27) = 1.778, *p* = 0.1936; Interaction between factors F (1, 27) = 0.7503, *p* = 0.3940. Females Treatment F (1, 23) = 8.886, *p* = 0.0067; Stress F (1, 23) = 0.2511, *p* = 0.6211; Interaction between factors F (1, 23) = 6.899 × 10^−5^, *p* = 0.9934). These results suggest that the increase in β-EP levels is dependent on the type of stress (forced swim stress) and the interaction between this type of stress and 3α,5α-THP treatment (Figure 6b). 

#### 3.3.3. 3α,5α-THP Altered MOP Receptors Differently, Depending on Sex and Type of Stress

β-endorphins exert their effect mainly on the Mu-opioid receptors [32]. We investigated the possibility that 3α,5α-THP-altered expression of the MOP receptors in the hypothalamus of the rats exposed to forced swim stress. The results showed a different effect induced by 3α,5α-THP, depending on sex. 

In male rats, 3α,5α-THP reduced hypothalamic MOP levels in absence of stress. The exposure to FSS did not affect MOP concentration, and the treatment with 3α,5α-THP did not change this condition (Treatment F (1, 35) = 15.29, *p* = 0.0004; Stress F (1, 35) = 5.498, *p* = 0.0248; Interaction between factors F (1, 35) = 1.920, *p* = 0.1746). In contrast, in female rats, 3α,5α-THP or FSS per se did not affect hypothalamic MOP levels, however the combination of the 3α,5α-THP treatment and the exposure to the forced swim stress induced an increase in the MOP receptor (Treatment F (1, 36) = 6.774, *p* = 0.0133; Stress F (1, 36) = 1.646, *p* = 0.2077; Interaction between factors F (1, 36) = 8.992, *p* = 0.0049) (Figure 7a).

3α,5α-THP administration showed a different effect when the animals were exposed to restraint stress. In fact, in both male and female rats, both restraint stress and 3α,5α-THP treatment decreased the concentration of MOP in the hypothalamus. However, the interaction between 3α,5α-THP and restraint stress induced an increase in MOP (Males Treatment F (1, 21) = 1.509, *p* = 0.2329; Stress F (1, 21) = 0.1261, *p* = 0.7261; Interaction between factors F (1, 21) = 40.62, *p* < 0.0001. Females Treatment F (1, 30) = 0.6257, *p* = 0.4352; Stress F (1, 30) = 1.024, *p* = 0.3197; Interaction between factors F (1, 30) = 53.49, *p* < 0.0001) (Figure 7b).

### 3.4. Experiment 3: FSS Behavioral Analysis Following Mu Opioid Receptors Inhibition with CTAP Prior to 3α,5α-THP Administration

#### 3.4.1. CTAP Administration Prior to 3α,5α-THP Treatment Prevented Submersion Behavior Induced by 3α,5α-THP in Male, but Not in Female Rats

To assess the hypothesis that 3α,5α-THP interaction with the opioid system induced the submersion behavior, we blocked the Mu-opioid receptors using the selective MOP antagonist CTAP at two different doses (0.5 and 2 mg/kg) prior to 3α,5α-THP treatment.

The behavioral analysis showed that, in male rats, treatment with CTAP at the higher dose (2 mg/kg, 30 min before 3α,5α-THP administration and 45 min before the FSS exposure) completely reversed the effect of 3α,5α-THP on submersion behavior, reducing the number of episodes (Treatment F (4, 37) = 7.780, *p* = 0.0001) (Figure 8a). This data strongly suggests that submersion behavior is caused by a deleterious interaction between 3α,5α-THP and the Mu-opioid system (Figure 8a).

In female rats, a similar trend was observed following CTAP (2 mg/kg) in conjunction with prior 3α,5α-THP treatment. However, although the statistical analysis resulted in an overall significant effect, no significant changes were detected with the multiple comparison test of both doses of CTAP on submersion behavior (n of episodes: Treatment F (4, 38) = 8.111, *p* < 0.0001). Nevertheless, the fact that the total number of episodes was clearly reduced and not different from the effects of vehicle alone suggests that CTAP regulates the 3α,5α-THP-induced effect on submersion behavior in female animals as well (Figure 8b).

#### 3.4.2. The Higher Dose of CTAP Prior to 3α,5α-THP Administration Reduced the Number of Immobility Episodes but Increased the Immobility Time in Male and Female Rats

Despite the lack of 3α,5α-THP effect on the number of episodes of immobility behavior observed in experiment 1 (Figure 2), 2 mg/kg CTAP prior to 3α,5α-THP administration reduced the number of immobility episodes in male rats (Treatment F (4, 37) = 5.711, *p* = 0.0011). The lower CTAP dose (0.5 mg/kg) did not change the 3α,5α-THP-induced increase in the total time for immobility in male rats. However, the higher CTAP dose (2 mg/kg) followed by 3α,5α-THP treatment induced a greater increase than 3α,5α-THP per se in this parameter (Treatment F (4, 34) = 21.57, *p* < 0.00010 (Figure 9a).

In female animals, CTAP (2 mg/kg prior to 3α,5α-THP administration) reversed the 3α,5α-THP-induced increase in the number of immobility episodes (Treatment F (4, 38) = 6.296, *p* = 0.0005). However, the same dose of CTAP increased the duration of immobility in female rats (Treatment F (4, 35) = 9.650, *p* < 0.0001) (Figure 9b).

#### 3.4.3. CTAP Administration Prior to 3α,5α-THP Treatment Did Not Affect the Body Position in the Water during Immobility

In male rats, CTAP prior to 3α,5α-THP administration did not alter the effect of 3α,5α-THP on the rats’ body posture in the water. In fact, 0.5 mg/kg CTAP resulted in an increase in α after 3α,5α-THP administration, compared to the control group (Treatment F (4, 37) = 4.727, *p* = 0.0035) (Figure 8c). 

In female rats, 3α,5α-THP per se increased α of the body position in the water; however, CTAP treatment prior to 3α,5α-THP administration did not induce any changes (Treatment F (4, 39) = 4.180, *p* = 0.0065) (Figure 10).

#### 3.4.4. CTAP Following 3α,5α-THP Treatment Reduced Swimming Time in Both Male and Female Rats

Consistent with the increase in immobility time, in male rats the lower dose of CTAP (0.5 mg/kg) did not change the 3α,5α-THP-induced decrease in the duration spent swimming. Moreover, the higher CTAP (2 mg/kg) dose prior to 3α,5α-THP administration enhanced this effect, further reducing the swimming total time (Treatment F (4, 36) = 14.70, *p* < 0.0001) (Figure 11a).

In female rats, the administration of the lower CTAP dose prior to 3α,5α-THP treatment did not affect swimming behavior; however, the higher dose resulted in a reduction similar to the one observed after 3α,5α-THP administration alone (Treatment F (4, 39) = 5.505, *p* = 0.00130) (Figure 11b).

#### 3.4.5. CTAP High Dose Administration Prior to 3α,5α-THP Treatment Reduced Climbing Behavior in Male Rats, but Did Not Change the 3α,5α-THP-Induced Effect in Female Rats

In male rats, 2 mg/kg CTAP decreased climbing behavior, both number of episodes (Treatment F (4, 36) = 2.886, *p* = 0.0359) and total time (Treatment F (4, 37) = 4.263, *p* = 0.0062) (Figure 12a). 

3α,5α-THP alone reduced both the number of episodes and time spent climbing in female rats. At the high dose, CTAP prior to 3α,5α-THP administration resulted in a similar effect, reducing both these parameters, while the low dose of CTAP did not induce any changes (n of episodes: Treatment F (4, 37) = 6.775, *p* = 0.0003; duration: Treatment F (4, 37) = 9.794, *p* < 0.0001) (Figure 12b). 

#### 3.4.6. CTAP Administration Prior to 3α,5α-THP Treatment Effect on Exploration

CTAP did not influence exploration behavior in male rats, similar to 3α,5α-THP treatment alone (number of episodes: Treatment F (4, 37) = 1.627, *p* = 0.1881; total time: Treatment F (4, 37) = 1.946, *p* = 0.1233) (Figure 13a). 

In female rats, CTAP prior to 3α,5α-THP administration reduced the number of exploration episodes at the low dose (Treatment F (4, 37) = 4.317, *p* = 0.0058) and the time spent exploring at both CTAP doses (Treatment F (4, 37) = 6.295, *p* = 0.0006) (Figure 13b).

## 4. Discussion

We previously demonstrated that 3α,5α-THP regulation of the HPA axis stress response is sex- and stressor-dependent [13]. In this study, we observed behavioral changes induced by 3α,5α-THP administration that might be the results of a synergic effect between 3α,5α-THP and the opioid system in animals exposed to forced swim stress. 3α,5α-THP administration produced a significant increase in involuntary submersion behavior during FSS. Moreover, we observed a difference in body posture in the water between the control and 3α,5α-THP groups. 3α,5α-THP administration caused the rats to adopt a vertical position more perpendicular to the water surface than rats who received vehicle. Specifically, 3α,5α-THP increased the angle (α) between the animals’ body and the water surface. Furthermore, the 3α,5α-THP-induced change in body posture positively correlated with the number of submersion episodes. In addition, the time spent immobile was increased in the animals treated with 3α,5α-THP, although the number of immobility episodes was not affected by 3α,5α-THP, again with no sex difference. 

It has previously been demonstrated that 3α,5α-THP administration at different doses (0.5, 1 and 2 mg/kg, IP) 30 min prior to FSS exposure decreased the duration of immobility [33]. This reduction in the time spent immobile was considered an anti-despaired or antidepressant-like effect of 3α,5α-THP. This discrepancy between Kristi’s study [33] and our results could be due to the difference in animal species (mice vs. rats), doses (0.5–2 mg/kg vs. 15 mg/kg), or timing (30 min vs. 15 min) used in this study. The dose used in this study (15 mg/kg) was chosen based on its efficacy in reducing CRF levels in our previous studies [12,13].

3α,5α-THP decreased time spent swimming in both male and female rats while climbing behavior was not affected by 3α,5α-THP administration. Finally, exploring behavior was not significantly affected by 3α,5α-THP in male rats; however, in female rats, 3α,5α-THP decreased both the number of episodes and the time spent exploring. Together, the data show that 3α,5α-THP impacts passive coping behaviors during FSS (submersion and immobility) in both male and female animals. In contrast, 3α,5α-THP did not influence active behaviors in male rats. Although 3α,5α-THP reduced swimming behavior in males and females, 3α,5α-THP did not change the other active behavior parameters (climbing and exploring) in male rats. In female rats, 3α,5α-THP did not affect climbing but reduced active coping behavior to explore the environment to find a way to escape. 

Since it has been demonstrated that the 3α,5α-THP dose of 15 mg/kg had no hypnotic effect [16] in rats, we did not expect such a dramatic behavioral sequelae in the 3α,5α-THP-treated group. However, the submersion, increase in immobility, and decrease in swimming suggest that 3α,5α-THP may have affected the animals’ locomotor or psychomotor activity. In the locomotor activity measurements, the results suggested that 3α,5α-THP alone did not alter locomotor activity in male and female rats. However, since the chamber used to measure the locomotor activity was smaller than a regular open field apparatus, we could not exclude the possibility that 3α,5α-THP induced an effect in locomotion. An open field test should be performed to address this concern. Therefore, it appears that 3α,5α-THP interacted with the effects of swim stress, indicating a potential deleterious behavior.

It is important to note that FSS is routinely used to screen drugs with antidepressant activity and that only in recent years has it been proposed to test measures of stress coping strategies rather than depression-like phenotypes [34]. It has been demonstrated that stress activates the opioid system, inducing the synthesis and release of different endogenous opioids [18]. Many brain structures that regulate physiological and behavioral components of the stress response, including cingulate cortex, amygdala, and periaqueductal gray, contain endogenous opioids that are released in response to stress and that are involved in regulating the HPA stress response, mood changes, and drug reward [18,35,36].

Our data showed that swim stress increased circulating β-EP levels in both male and female rats. Interestingly, the combination of 3α,5α-THP and FSS enhanced this effect, suggesting that the participation of the opioid system is responsible for the behavioral outcome we observed. The increase in β-EP might account for the increase in submersion behavior observed in this study. Moreover, β-EP levels did not change after restraint stress (RS) exposure, suggesting that the activation of the HPA axis stress response is stressor-dependent, as we previously showed [13]. One important factor to consider is the different temperature the animals faced from the two different stressors (the water in the swim stress at body temperature vs. the room temperature during restraint stress at 21 °C); we cannot exclude the possibility that this could be an important variable that influenced the different results in our experiment.

Administration of the selective MOP antagonist CTAP at two different doses prior to 3α,5α-THP administration changed the behavioral outcome during FSS. In male rats, the high dose of CTAP completely reversed the 3α,5α-THP effect on submersion behavior, suggesting, once again, that the contribution of the Mu-opioid receptor system resulted in this specific behavioral effect. In female rats, CTAP led to the same trend observed in male rats, but it was not statistically significant. Overall, pharmacological manipulation of MOPS with CTAP seemed to produce an effect on specific behaviors (reducing submersion, increasing immobility time, decreasing climbing behaviors). 

Interactions between neurosteroids and the opioid system have previously been reported. In fact, previous studies in male mice showed that levels of progesterone directly increase sigma1 receptor effects [37,38]. Other studies showed that DHEA sulfate acted as a sigma1 receptor agonist [39,40]. In female rats, the increase in estradiol and progesterone during pregnancy activate the kappa-opiate receptor analgesic system [41]. Furthermore, β-EP serum levels had been related to estrous cycle in rats, suggesting that estradiol and progesterone control β-EP synthesis and release [42]. Human studies have shown that β-EP levels change during the menstrual cycle, suggesting that β-EP levels seem to be related to ovulatory function [43]. Specifically, β-EP increase during the periovulatory phase appears to be correlated with the typical increase in plasma estradiol levels during this phase [44]. Moreover, a decrease β-EP levels has been shown in post-menopausal women: the decrease in β-EP levels during menopause has been associate with hot flashes episodes, sweating [45] and to the development of mood, behavior and nociceptive alterations of post-menopausal syndrome [46].

We observed no sex differences in β-EP levels after 3α,5α-THP administration and swim stress exposure; however, MOP expression showed a sex dependent regulation. In fact, in male rats, both 3α,5α-THP treatment per se or swim stress alone induced a decrease in hypothalamic MOP expression, but the combination of 3α,5α-THP + FSS did not produce any effect. In contrast, in female animals, we did not observe any change due to 3α,5α-THP treatment or swim stress; however, the combination of 3α,5α-THP + FSS resulted in an increase in hypothalamic MOP levels. It is possible that the timing and/or the brain region studied here were not related to the behavioral effects we observed. For example, it might be interesting to analyze MOP expression in brain regions involved in decision making processes, such as the prefrontal cortex and projecting targets. In fact, it has been demonstrated that a possible mechanism responsible for the cognitive function impairment due to opioid drug use is the suppression of cortical interneuron spiking by mu-opioid receptors; this effect results in an increase in glutamate that led to a disorganization in control and decision processes [47]. Further studies of the mechanisms underlying the deleterious interactions of 3α,5α-THP and the opiate system are warranted.

This is the first study that has shown a possible deleterious behavior due to the interaction between 3α,5α-THP and the opioid system in the context of FSS. Further studies are needed to understand this phenomenon; this study is an indirect observation, and further direct investigation of 3α,5α-THP and opiate interactions are required. It would be interesting, for example, to explore the effect of the combination of 3α,5α-THP and opioid agonists (such as fentanyl) at sub-threshold doses for multiple behavioral effects. 

These findings might be important to better understand the limitations of 3α,5α-THP in the treatment of neuropsychiatric disease. Several neuropsychiatric disorders, such as major depression, post-traumatic stress disorder, and substance use disorders share a dysregulation of the HPA axis as well as an imbalance between the glutamatergic excitatory inputs and GABAergic inhibitory transmission across brain [4]. The rationale for 3α,5α-THP treatment of substance use disorders, such as cocaine and alcohol, is restoring the normal HPA axis response to stress as well as the glutamatergic/GABAergic signaling imbalance in the brain. Indeed, the efficacy of 3α,5α-THP administration to reduce cocaine reinstatement was observed in cocaine-dependent animals [48,49], and female rats were more sensitive to this effect [50,51]. Other findings showed that pregnenolone and its GABAergic metabolites regulate alcohol motivation by reducing alcohol seeking and intake in rodents [52,53,54]. These results were confirmed in clinical studies in patients with history of cocaine and alcohol use disorders [55,56]. However, the data presented in this study suggest a negative interaction between 3α,5α-THP and the opioid system that might result in a deleterious behavior. Further studies will be needed to validate this theory and better understand the mechanism underlying this behavior.

## 5. Conclusions

This study suggests that 3α,5α-THP has a deleterious effect combined with FSS, which together induced an exacerbation of the activation of the opioid system, resulting in submersion behavior. This effect was prevented with the Mu-opioid receptor antagonist CTAP in rats. These results might be an important contribution to the development of new treatments for patients with neuropsychiatric disorders that involve HPA axis activation. The data, in fact, suggest that 3α,5α-THP/opiate system interactions may lead to negative behavioral outcomes. Future studies are needed to better support this hypothesis. Forthcoming perspectives to confirm our theory should be activating the opioid system using an opioid agonist concurrently with 3α,5α-THP administration, to evaluate the level of anesthesia and the interaction between the two drugs. 

## Figures and Tables

**Figure 1 biomolecules-13-01205-f001:**
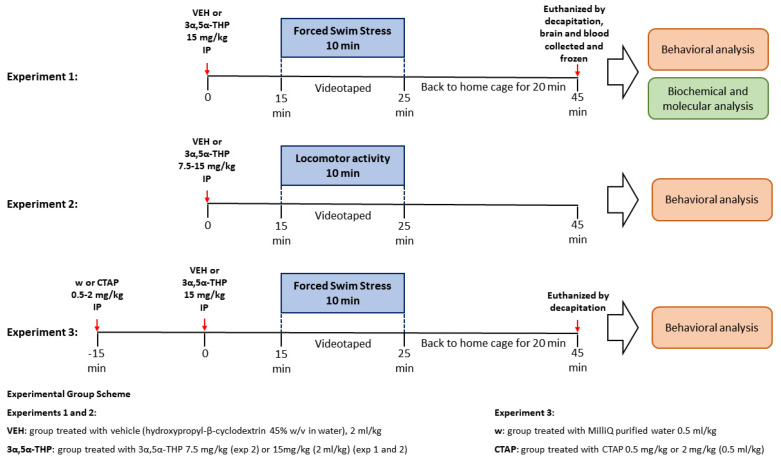
Experiments timeline.

**Figure 2 biomolecules-13-01205-f002:**
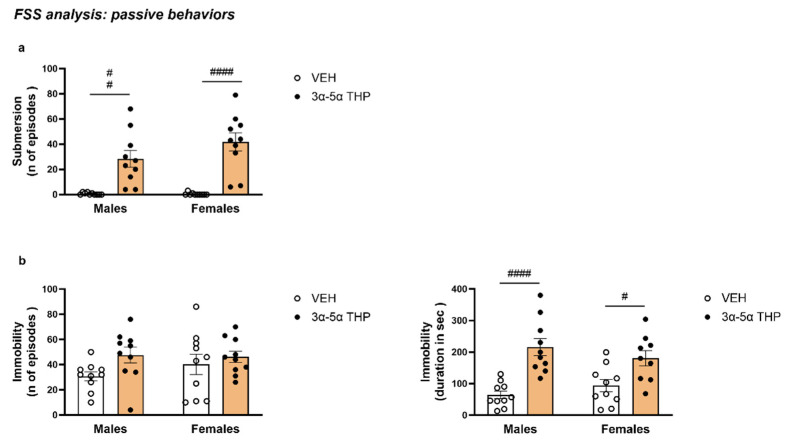
3α,5α-THP increased passive behavior during FSS, inducing submersion behavior in male and female rats. (**a**) In both male and female animals, 3α,5α-THP administration increased the number of submersion episodes (Males: VEH = 0.6 ± 0.27 vs. 3α,5α-THP = 28.4 ± 6.58, Treatment effect *p* < 0.01; Females: VEH = 0.4 ± 0.31 vs. 3α,5α-THP = 41.8 ± 7.13, Treatment effect *p* < 0.0001). (**b**) Moreover, 3α,5α-THP treatment increased immobility time (Males: VEH = 64.54 ± 12.18 vs. 3α,5α-THP = 216.3 ± 26.97, Treatment effect *p* < 0.0001; Females: VEH = 93.36 ± 19.33 vs. 3α,5α-THP = 180.4 ± 24.61, Treatment effect *p* < 0.05) but not the number of episodes (Males: VEH = 30.8 ± 3.53 vs. 3α,5α-THP = 47.7 ± 6.27, n.s.; Females: VEH = 40.3 ± 8.07 vs. 3α,5α-THP = 46.2 ± 4.54, n.s.). The data suggest that 3α,5α-THP increases passive behaviors during FSS. Significant effects were found using two-way ANOVA, Treatment effect ^####^
*p* < 0.0001, ^##^
*p* < 0.01, ^#^
*p* < 0.05. Abbreviations: VEH = rats treated with vehicle; 3α,5α-THP = rats treated with 3α,5α-THP; n.s. = not significant.

**Figure 3 biomolecules-13-01205-f003:**
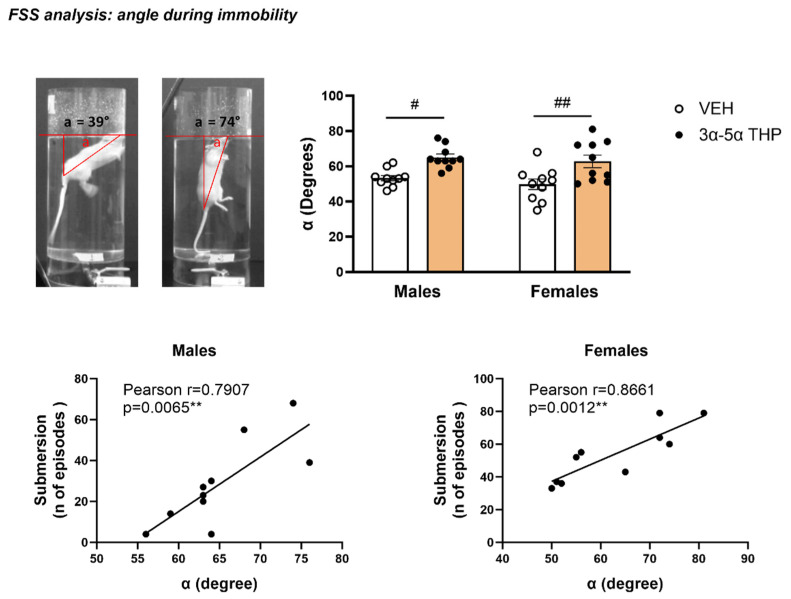
3α,5α-THP changed the animals’ body position in the water, and this effect was positively correlated with submersion behavior. In both male and female animals, 3α,5α-THP treatment altered body posture in the water during FSS, increasing the angle (α) between the animal’s body and the water surface (Males: VEH = 53.3 ± 1.59 vs. 3α,5α-THP = 65 ± 1.95, Treatment effect *p* < 0.01; Females: VEH = 49.8 ± 3 vs. 3α,5α-THP = 62.8 ± 3.59, Treatment effect *p* < 0.01). This increase is positively correlated with the number of submersion episodes (Males: Pearson r = 0.907 *p* = 0.0065; Females: Pearson r = 0.8661, *p* = 0.0012), suggesting that the change in body posture is predictive of submersion behavior. Effects were analyzed using two-way ANOVA, Treatment effect ^##^
*p* < 0.01, ^#^
*p* < 0.05. Significant correlations were calculated using Pearson’s test and linear regression analysis, *** p* < 0.01. Abbreviations: VEH = rats treated with vehicle; 3α,5α-THP = rats treated with 3α,5α-THP.

**Figure 4 biomolecules-13-01205-f004:**
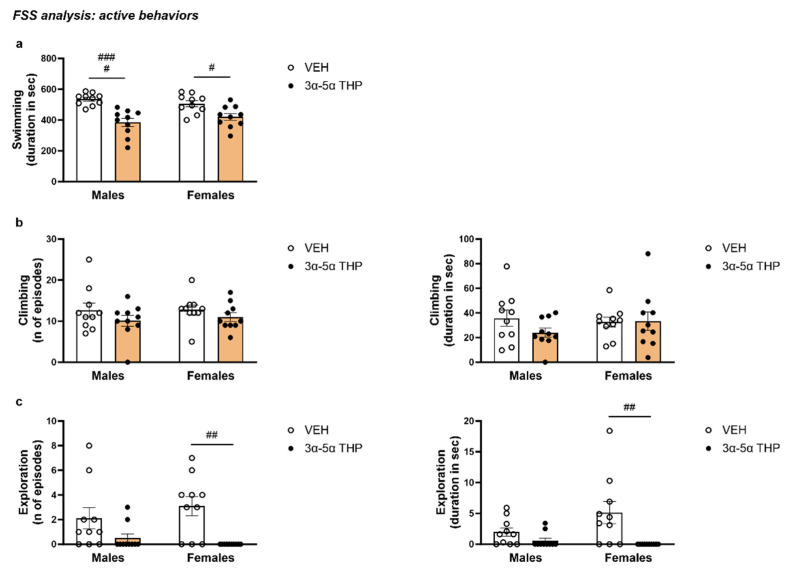
3α,5α-THP effects on active behaviors. (**a**) 3α,5α-THP decreased the total time the animals spent swimming (Males: VEH = 535.5 ± 12.18 vs. 3α,5α-THP = 383.7 ± 26.97, Treatment effect *p* < 0.0001; Females: VEH = 506.6 ± 19.33 vs. 3α,5α-THP = 421.3 ± 22.07, Treatment effect *p* < 0.05). (**b**) 3α,5α-THP administration did not alter climbing behavior in male or female animals (number of episodes: Males: VEH = 12.7 ± 1.67 vs. 3α,5α-THP = 10.1 ± 1.33, n.s.; Females VEH = 12.8 ± 1.14 vs. 3α,5α-THP = 11 ± 1.03; total time Males VEH = 35.79 ± 6.48 vs. 3α,5α-THP = 24.07 ± 3.77; Females VEH = 32.58 ± 4.03 vs. 3α,5α-THP = 33.38 ± 7.44, n.s.). (**c**) In female rats, 3α,5α-THP reduced the number of episodes (VEH = 3.1 ± 0.78 vs. 3α,5α-THP = 0 ± 0, Treatment effect *p* < 0.01) and total time spent exploring the environment (VEH= 5.14 ± 1.8 vs. 3α,5α-THP = 0 ± 0, Treatment effect *p* < 0.01), while in male rats we did not observe any change in the exploration behavior (n of episodes VEH = 2.1 ± 0.86 vs. 3α,5α-THP = 0.5 ± 0.34, n.s.; duration VEH = 1.98 ± 0.68 vs. 3α,5α-THP = 0.59 ± 0.4, n.s.). Significant effect was found using Two-way ANOVA, Treatment effect ^####^
*p* < 0.0001, ^##^
*p* < 0.01, ^#^
*p* < 0.05. Abbreviations: VEH = rats treated with vehicle; 3α,5α-THP = rats treated with 3α,5α-THP.

**Figure 5 biomolecules-13-01205-f005:**
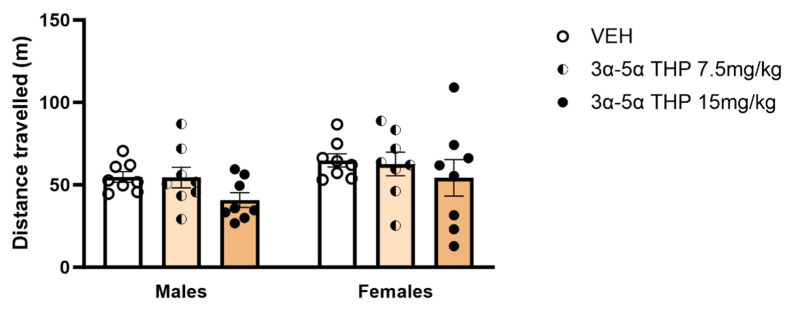
3α,5α-THP did not influence the distance traveled during locomotor activity measurement. 3α,5α-THP (7.5 and 15 mg/kg) did not affect the distance traveled during locomotor activity measurement, in male (VEH = 54.73 ± 3.178; 3α,5α-THP 7.5 mg/kg = 54.43 ± 6.302; 3α,5α-THP 15 mg/kg = 40.69 ± 4.435, n.s.) or female animals (VEH = 64.82 ± 4.031; 3α,5α-THP 7.5 mg/kg = 62.62 ± 7.162; 3α,5α-THP 15 mg/kg = 54.27 ± 11.03, n.s.). Two-way ANOVA analysis. Abbreviations: VEH = rats treated with vehicle; 3α,5α-THP = rats treated with 3α,5α-THP.

**Figure 6 biomolecules-13-01205-f006:**
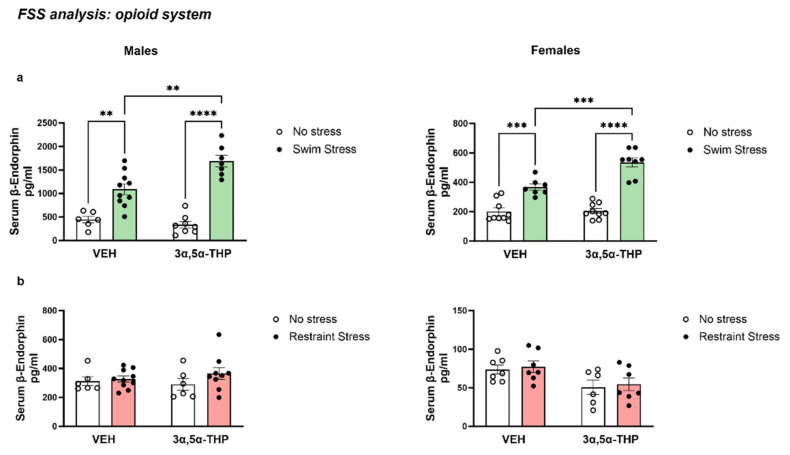
3α,5α-THP treatment increased serum β-endorphins concentrations when combined with forced swim stress but not restraint stress. (**a**) The exposure to forced swim stress increased serum β-endorphins levels in both male and female rats. Moreover, 3α,5α-THP administration enhanced this effect, and the animals treated with 3α,5α-THP showed a greater increase in β-endorphins levels than the vehicle group in both male (VEH No stress= 445.2 ± 68.62 vs. VEH Swim Stress = 1093 ± 116.2, *p* < 0.01; 3α,5α-THP No stress = 336.4 ± 70.24 vs. 3α,5α-THP Swim Stress = 1688 ± 123.7, *p* < 0.0001; VEH Swim Stress = 1093 ± 116.2 vs. 3α,5α-THP Swim Stress = 1688 ± 123.7, *p* < 0.01) and female rats (VEH No stress = 200.8 ± 24.06 vs. VEH Swim Stress = 366.8 ± 21.33, *p* < 0.001; 3α,5α-THP No stress = 204.8 ± 17.5 vs. 3α,5α-THP Swim Stress = 535.2 ± 31.83, *p* < 0.0001; VEH Swim Stress = 200.8 ± 24.06 vs. 3α,5α-THP Swim Stress = 535.2 ± 31.83, *p* < 0.001). (**b**) This effect is dependent upon swim stress; in fact, we did not observe any changes in β-endorphin levels following restraint stress per se or after restraint stress combined with 3α,5α-THP treatment in male (VEH No stress = 311.6 ± 30.3 vs. VEH Swim Stress = 327.7 ± 20.48, n.s.; 3α,5α-THP No stress = 290.1 ± 40.61 vs. 3α,5α-THP Restraint Stress = 366 ± 41.04, n.s.) and female animals (VEH No stress = 73.5 ± 5.8 vs. VEH Swim Stress = 77.287 ± 7.41, n.s.; 3α,5α-THP No stress = 50.63 ± 9.16 vs. 3α,5α-THP Restraint Stress = 54.52 ± 8.19, n.s.). A significant effect was found using two-way ANOVA, following Tukey’s HSD test **** *p* < 0.0001, *** *p* < 0.001, ** *p* < 0.01. Abbreviations: VEH = rats treated with vehicle; 3α,5α-THP = rats treated with 3α,5α-THP.

**Figure 7 biomolecules-13-01205-f007:**
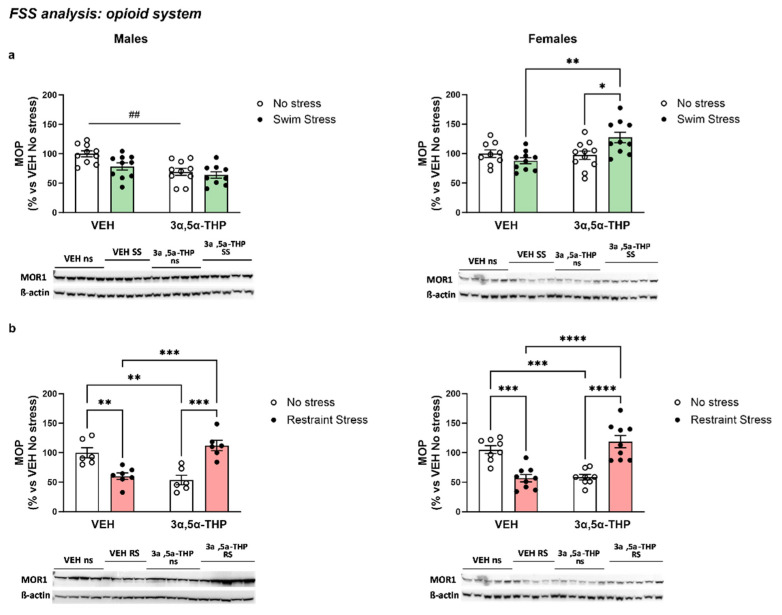
3α,5α-THP induced different changes in hypothalamic MOP depending on sex and type of stress. (**a**) In male rats, swim stress did not significantly change MOP (VEH swim stress = 78.48 ± 6.2 vs. VEH no stress, n.s.). 3α,5α-THP administration reduced MOP in the absence of swim stress (3α,5α-THP no stress = 69.44 ± 6.01% vs. VEH no stress, Treatment effect *p* < 0.01). The combination of swim stress and 3α,5α-THP did not produce a significant effect (3α,5α-THP swim stress = 63.91 ± 5.72 vs. VEH no stress, n.s.). In female rats, swim stress (VEH swim stress = 87.9 ± 5.06 vs. VEH no stress, n.s.) or 3α,5α-THP (3α,5α-THP no stress = 97.21 ± 6.99 vs. VEH no stress, n.s.) per se did not induce significant changes in hypothalamic MOP. However, 3α,5α-THP administration followed by swim stress induced a significant increase in MOP (3α,5α-THP swim stress = 127.4 ± 8.94 vs. 3α,5α-THP no stress = 97.21 ± 6.99, *p* < 0.05; vs. VEH swim stress = 87.9 ± 5.06, *p* < 0.01). (**b**) Following restraint stress, MOP concentration reduced in male (VEH restraint stress = 60.33 ± 5.55 vs. VEH no stress, *p* < 0.01) and female rats (VEH restraint stress = 56.78 ± 6.15 vs. VEH no stress, *p* < 0.001). 3α,5α-THP administration reduced MOP in absence of stress in male (3α,5α-THP no stress = 53.6 ± 8.03 vs. VEH no stress, *p* < 0.01) and female rats (3α,5α-THP no stress = 58.42 ± 4.63 vs. VEH no stress, *p* < 0.001). Moreover, the combination of 3α,5α-THP and restraint stress induced an increase in hypothalamic MOP in both male (3α,5α-THP swim stress = 112.2 ± 8.76 vs. 3α,5α-THP no stress = 53.6 ± 8.03, *p* < 0.001; vs. VEH restraint stress = 60.33 ± 5.55, *p* < 0.001) and female animals (3α,5α-THP swim stress = 118.7 ± 10.31 vs. 3α,5α-THP no stress = 58.42 ± 4.63, *p* < 0.0001; vs. VEH restraint stress = 56.78 ± 6.15, *p* < 0.0001). Significant effects were found using two-way ANOVA ^##^ *p* < 0.01, following Tukey’s HSD test **** *p* < 0.0001, *** *p* < 0.001, ** *p* < 0.01, * *p* < 0.05. Data are represented as % ± SEM vs. VEH no stress. All % values are calculated vs. their respective VEH no stress control group. Abbreviations: VEH = rats treated with vehicle; 3α,5α-THP = rats treated with 3α,5α-THP.

**Figure 8 biomolecules-13-01205-f008:**
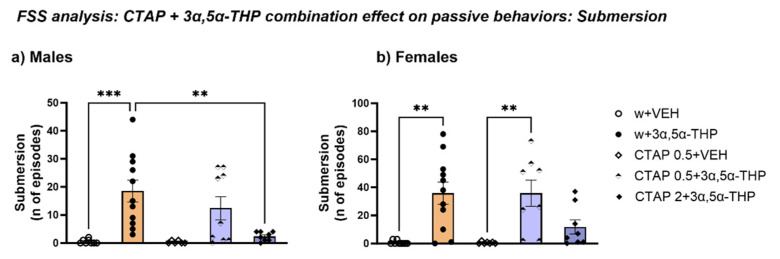
CTAP administration prior to 3α,5α-THP treatment reversed 3α,5α-THP-induced submersion behavior in male rats. (**a**) At the higher dose (2 mg/kg), CTAP administration 30 min before 3α,5α-THP injection prevented the effect on submersion behavior induced by 3α,5α-THP alone in male rats. In fact, the number of submersion episodes was reduced by CTAP (2 mg/kg) in the animals treated with 3α,5α-THP (w + VEH = 0.5 ± 0.27 vs. w + 3α,5α-THP = 18.5 ± 3.9, *p* < 0.001; w + 3α,5α-THP = 18.5 ± 3. vs. CTAP 2 + 3α,5α-THP = 2.38 ± 0.57, *p* < 0.01). (**b**) In female rats, the higher CTAP dose (2 mg/kg) following 3α,5α-THP treatment had no significant effect on the number of submersion episodes, although the total number of episodes was clearly reduced and not different from the effects of vehicle alone (w + VEH = 0.67 ± 0.44 vs. w + 3α,5α-THP = 36.91 ± 7.9, *p* < 0.01; CTAP + VEH = 0.57 ± 0.3 vs. CTAP 0.5 + 3α,5α-THP = 35.88 ± 9.3, *p* < 0.01; vs. CTAP 2 + 3α,5α-THP = 11.88 ± 5.12, n.s.). Significant effects were found using one-way ANOVA *** *p* < 0.001, ** *p* < 0.01. Abbreviations: w = rats treated first with water; CTAP = rats treated first with 0.5 or 2 mg/kg CTAP; VEH = rats received the second injection with vehicle; 3α,5α-THP = rats received the second injection with 3α,5α-THP.

**Figure 9 biomolecules-13-01205-f009:**
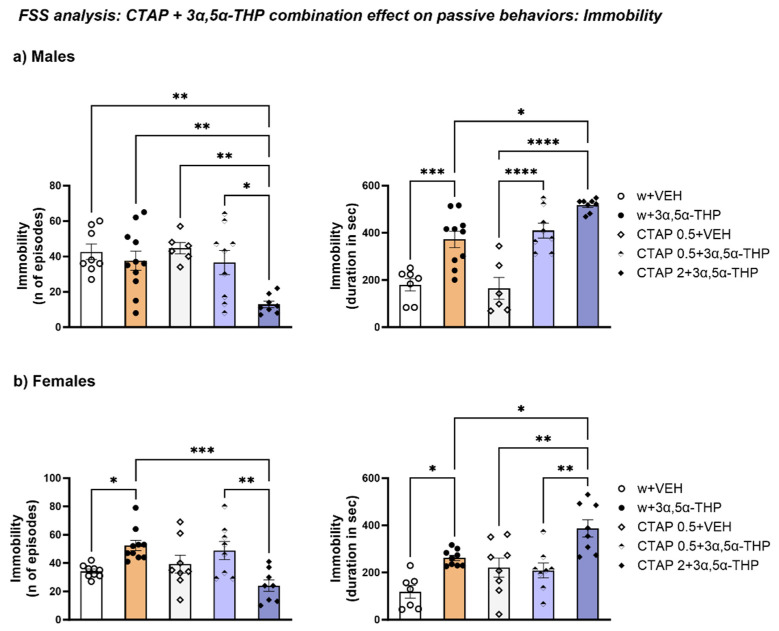
CTAP administration prior to 3α,5α-THP treatment effect on immobility. (**a**) In male rats, 2 mg/kg CTAP prior to 3α,5α-THP administration reduced the number of immobility episodes (CTAP 2 + 3α,5α-THP = 12.88 ± 1.86 vs. w + VEH = 42.63 ± 4.42, *p* < 0.01; vs. w + 3α,5α-THP = 37.64 ± 5.4, *p* < 0.01; vs. CTAP + VEH = 34 ± 3.18, *p* < 0.01; vs. CTAP 0.5 + 3α,5α-THP = 36.56 ± 6.83, *p* < 0.05). However, the combination CTAP + 3α,5α-THP did not change the 3α,5α-THP-induced increase in the total time spent immobile; moreover, at the higher dose (2 mg/kg), CTAP administration following 3α,5α-THP injection enhanced the effect of 3α,5α-THP alone in male animals (w + VEH = 179.8 ± 25.9 vs. w + 3α,5α-THP = 372.6 ± 34.75, *p* < 0.001; CTAP + VEH = 165 ± 46.17 vs. CTAP 0.5 + 3α,5α-THP = 409.7 ± 31.47, *p* < 0.0001; vs. CTAP 2 + 3α,5α-THP = 517.2 ± 9.79, *p* < 0.0001; w + 3α,5α-THP = 372.6 ± 34.75 vs. CTAP 2 + 3α,5α-THP = 517.2 ± 9.79, *p* < 0.05). (**b**) As observed in males, in female rats 2 mg/kg CTAP prior to 3α,5α-THP administration reduced the number of immobility episodes (w + VEH = 34.2 ± 1.49 vs. w + 3α,5α-THP = 52.4 ± 3.61, *p* < 0.05; CTAP 2 + 3α,5α-THP = 24.13 ± 4 vs. CTAP 0.5 + 3α,5α-THP = 48.88 ± 6.4, *p* < 0.01; vs. w + 3α,5α-THP = 52.4 ± 3.61, *p* < 0.001), but increased the immobility time, compared to the 3α,5α-THP-treated group (w + VEH = 118.6 ± 26.7 vs. w + 3α,5α-THP = 262 ± 11.26, *p* < 0.05; CTAP + VEH = 221.4 ± 40.7 vs. CTAP 0.5 + 3α,5α-THP = 208.6 ± 31.7, *p* < 0.01; CTAP 2 + 3α,5α-THP = 387.3 ± 36.84 vs. CTAP 0.5 + 3α,5α-THP = 208.6 ± 31.7, *p* < 0.01; vs. w + 3α,5α-THP = 262 ± 11.26, *p* < 0.5). Significant effects were found using one-way ANOVA **** *p* < 0.0001, *** *p* < 0.001, ** *p* < 0.01, * *p* < 0.05. Abbreviations: w = rats treated first with water; CTAP = rats treated first with CTAP 0.5 or 2 mg/kg; VEH = rats received the second injection with vehicle; 3α,5α-THP = rats received the second injection with 3α,5α-THP.

**Figure 10 biomolecules-13-01205-f010:**
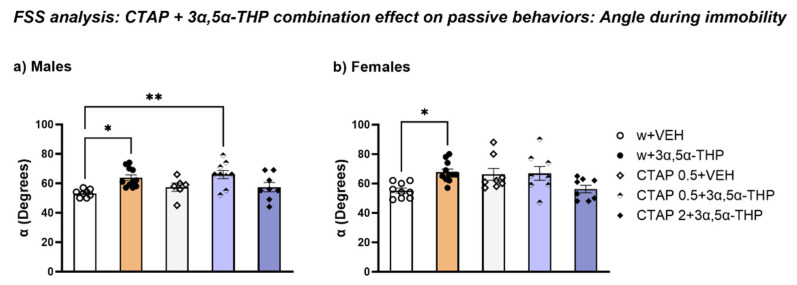
CTAP administration prior to 3α,5α-THP treatment did not affect body posture in the water. (**a**) 0.5 mg/kg CTAP following 3α,5α-THP administration resulted in the same effect as 3α,5α-THP alone, increasing the angle of body posture in the water in male rats (w + VEH = 53.25 ± 0.9 vs. w + 3α,5α-THP = 63.73 ± 1.96, *p* < 0.05; vs. CTAP 0.5 + 3α,5α-THP = 66 ± 2.83, *p* < 0.01). Significant effects were found using one-way ANOVA. Abbreviations: w = rats treated first with water; CTAP = rats treated first with CTAP 0.5 or 2 mg/kg; VEH = rats received the second injection with vehicle; 3α,5α-THP = rats received the second injection with 3α,5α-THP. (**b**) While 3α,5α-THP per se induced an increase, both doses of CTAP prior to 3α,5α-THP administration did not induced any changes in the position of the body into the water in female rats (w + VEH = 55.22 ± 1.7 vs. w + 3α,5α-THP = 67.82 ± 2.1, *p* < 0.05). Significant effects were found using one-way ANOVA ** *p* < 0.01, * *p* < 0.05. Abbreviations: w = rats treated first with water; CTAP = rats treated first with 0.5 or 2 mg/kg CTAP; VEH = rats received the second injection with vehicle; 3α,5α-THP = rats received the second injection with 3α,5α-THP.

**Figure 11 biomolecules-13-01205-f011:**
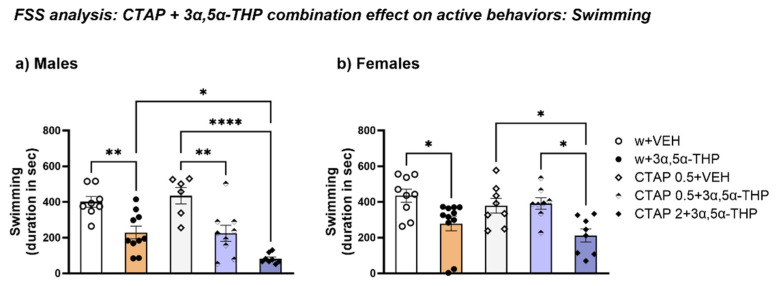
CTAP administration prior to 3α,5α-THP treatment reduced swimming behavior in male rats. (**a**) As observed after 3α,5α-THP administration, both doses of CTAP reduced the time spent swimming in male rats. Moreover, at the higher dose (2 mg/kg) the combination with CTAP produced an effect that was greater than the 3α,5α-THP reduction per se (w + VEH = 400.7 ± 29.74 vs. w + 3α,5α-THP = 229 ± 35.66, *p* < 0.01; CTAP + VEH = 435 ± 46.17 vs. CTAP 0.5 + 3α,5α-THP = 225 ± 44.48, *p* < 0.01; vs. CTAP 2 + 3α,5α-THP = 82.81 ± 9.78, *p* < 0.0001; w + 3α,5α-THP = 229 ± 35.66 vs. CTAP 2 + 3α,5α-THP = 82.81 ± 9.78, *p* < 0.05). (**b**) In female rats, 3α,5α-THP administration reduced the time spent swimming during FSS (w + VEH = 435.1 ± 36.8 vs. w + 3α,5α-THP = 278.9 ± 40.65, *p* < 0.05). The lower dose of CTAP combined with 3α,5α-THP administration did not affect the swimming behavior, however, 2 mg/kg CTAP reduced it (CTAP + VEH = 378.6 ± 40.7 vs. CTAP 0.5 = 3α,5α-THP = 391.4 ± 31.7, n.s.; vs. CTAP 2 + 3α,5α-THP = 212.7 ± 36.8, *p* < 0.05). Significant effects were found using one-way ANOVA **** *p* < 0.0001, ** *p* < 0.01, * *p* < 0.05. Abbreviations: w = rats treated first with water; CTAP = rats treated first with 0.5 or 2 mg/kg CTAP; VEH = rats received the second injection with vehicle; 3α,5α-THP = rats received the second injection with 3α,5α-THP.

**Figure 12 biomolecules-13-01205-f012:**
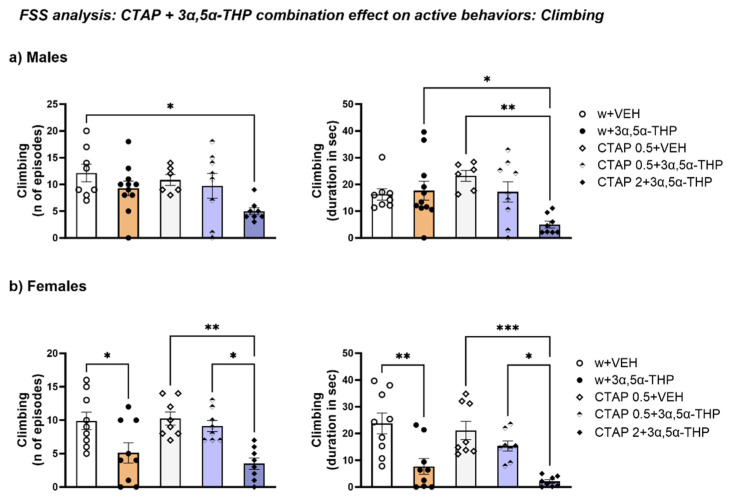
CTAP administration prior to 3α,5α-THP treatment reduced climbing behavior in male and female rats. (**a**) 2 mg/kg CTAP following 3α,5α-THP administration reduced the number of episodes (w + VEH = 12.13 ± 1.65 vs. CTAP 2 + 3α,5α-THP = 5 ± 0.66, *p* < 0.05) and the total time spent climbing in male rats (CTAP 2 + 3α,5α-THP = 4.99 ± 1.28 vs. w + 3α,5α-THP = 17.66 ± 3.5, *p* < 0.05; vs. CTAP + VEH = 23.2 ± 2.07, *p* < 0.01). (**b**) Similar results were observed for climbing behavior in female rats. 3α,5α-THP administration reduced number of episodes and time spent climbing (n of episodes: w + VEH = 9.89 ± 1.32 vs. w + 3α,5α-THP = 5.11 ± 1.52, *p* < 0.05; duration: w + VEH = 23.8 ± 3.91 vs. w + 3α,5α-THP = 7.73 ± 2.97, *p* < 0.01); 0.5 mg/kg CTAP following 3α,5α-THP administration did not change the number of episodes or the time spent climbing, however the higher dose of CTAP reduced those parameters in female rats (n of episodes: CTAP + VEH = 10.25 ± 0.98 vs. CTAP 0.5 + 3α,5α-THP = 9.13 ± 0.83, n.s.; vs. CTAP 2 + 3α,5α-THP = 3.5 ± 0.87, *p* < 0.01; duration: CTAP + VEH = 21.15 ± 3.43 vs. CTAP 0.5 + 3α,5α-THP = 15.3 ± 1.9, n.s.; vs. CTAP 2 + 3α,5α-THP = 2.06 ± 0.68, *p* < 0.001). Significant effects were found using one-way ANOVA *** *p* < 0.001, ** *p* < 0.01, * *p* < 0.05. Abbreviations: w = rats treated first with water; CTAP = rats treated first with CTAP 0.5 or 2 mg/kg; VEH = rats received the second injection with vehicle; 3α,5α-THP = rats received the second injection with 3α,5α-THP.

**Figure 13 biomolecules-13-01205-f013:**
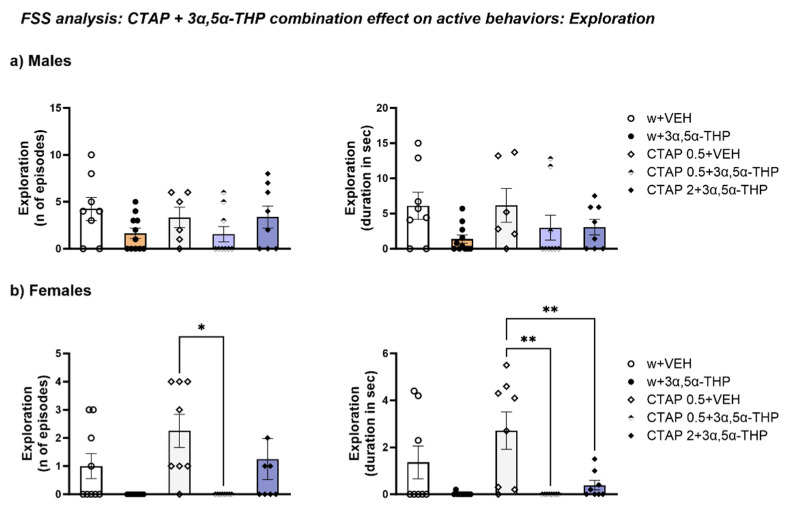
CTAP administration did not influence exploration behavior. (**a**) In male rats, CTAP did not induce any significant change in exploration behavior. (**b**) Both 0.5 mg/kg CTAP and 3α,5α-THP alone reduced exploration behavior in female rats. However, only 0.5 mg/kg CTAP followed by 3α,5α-THP administration resulted in a statistically significant decrease in the number of episodes (CTAP + VEH = 2.25 ± 0.59 vs. CTAP 0.5 + 3α,5α-THP = 0 ± 0, *p* < 0.05) and both doses of CTAP following 3α,5α-THP treatment in the time spent exploring (CTAP + VEH = 2.7 ± 0.79 vs. CTAP 0.5 + 3α,5α-THP = 0 ± 0, *p* < 0.01; vs. CTAP 2 + 3α,5α-THP = 0.39 ± 0.2, *p* < 0.01). Significant effects were found using one-way ANOVA ** *p* < 0.01, * *p* < 0.05. Abbreviations: w = rats treated first with water; CTAP = rats treated first with CTAP 0.5 or 2 mg/kg; VEH = rats received the second injection with vehicle; 3α,5α-THP = rats received the second injection with 3α,5α-THP.

## Data Availability

Not applicable.

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
