# Peer review of "Deleterious Interaction between the Neurosteroid (3α,5α)3-Hydroxypregnan-20-One (3α,5α-THP) and the Mu-Opioid System Activation during Forced Swim Stress in Rats"

_biomolecules, 2023, doi:10.3390/biom13081205_

Round 1
Reviewer 1 Report
The study by Boero et al reports on the life threatening effect in stress situation of the FDA approved neurosteroid allopregnanolone ( 3a,5a)3-hydroxy-pregnan-20-one or 3a,5a-THP) recently introduced into clinic for the treatment of post-partum depression. Allopregnanolone (3a,5a-THP) was also tested recently and showed positive effects in rodent models of the cocain-withdrawal syndrome. Effect (passive submersion under the water in vertical position in 20cm –diameter water tank) was detected now in a rat model of forced swim stress (FSS) under 3a,5a-THP (15mg/kg, i.p.). Some, but not all, measured parameters of this – for rats - abnormal behavior were sensitive to the BBB permeable mu-opioid receptor (MOP) antagonist CTAP. Using ELISA and immunoblotting, the authors show increase in b-endorphin and MOP levels after FSS and/or 3a,5a-THP (i.p.) with sex-specific variations (commenting paragraphs in introduction and discussion are identical and should be reformulated). The paper has numerous typos and presents very hard reading. I suggest to present findings which were CTAP-sensitive and those which were not CTAP-dependent separately e.g. in a table instead of numerous bar histograms which are not properly labelled. The finding that i.p. injection of water (vehicle) significantly increases b-endorphin level as well (not only FSS) should be mentioned in the discussion. All known types of stress which will upregulate b-endorphin should be listed. Minimal concentration of 3a,5a-THP combined with minimal concentration of fentanyl (synthetic MOP agonist) to produce LOR (loss of righting reflex) in rat should be determined or a paper cited where it was already done. My specific comments and suggestions for improvements are listed below. In general, I want to emphasize the importance of the finding and high time to see these data published.
Major concerns
11) Allopregnanolone is not only an anxiolytic, anticonvulsant and antidepressant drug, as authors mention in their introduction (lines 57-58), but also a general anesthetic (it can produce loss of consciousness or “hypnosis” in animal experiment and spinal areflexy). Supra-synergistic action between etomidate and the neurosteroid 5b-pregnan-3a-ol-20one, which binds to the same site on GABAAR as 3a,5a-THP, was reported in BJP 2014 by Li et al (PMID 25117207). Neurosteroid was injected i.p. in this study at 1mg/kg. Therefore, the combined effect of MOP agonist fentanyl and 3a,5a-THP needs to be studied. Best will be an EEG recording to estimate level of anesthesia.
22) According to the last IUPHAR classification opioid receptors are called MOP, DOP, KOP and NOP (Cox et al, “Challenges for opioid receptor nomenclature: IUPHAR Review 9”, BJP, 2015, 172(2), pp: 317-23)
Minor concerns
1) Lines 98,99. Last sentence of introduction is awkward and incorrect. Why opioid use disorder? Cocain and alcohol use disorders? Please introduce here your citations 40, 41,42,43,44-46. Same for the last sentence of discussion.
2) Line 40: Please introduce citations for “Postpartum depression”.
3) Use standard nomenclature for abbreviations, e.g. i.p. instead of IP and a.m. instead of AM.
4) “Euthanasie” generally means killing by overdosing general anaesthetics. Please provide more details on your procedures.
5) Fig.1 lowest panel. How can behavioral analysis can be done after euthanization?
6) Lines 310, 323 defects in punctuation in 3a,5a-THP
7) Fig 8 shows no labelling of bars.
8) Line 504. “..CTAP increased the time spent during immobility..” Spent in immobility?
9) Space is often missing between drug and dose
10) Line 517: ..in female rats CTAP2mg/kg… reduced the 3a,5a-THP-induced increased in immobility.."
11) Figures: symbol labelling is misleading. W+VEH is separated by 2 days injections of water (vehicle: VEH) . Authors should also show whether b-endorphin release was similar after repeated injection of VEH or showed habituation. Scheme of W, VEH injection should be added to the Fig.1.
12) Page 549. Behavioral excitation (evident from experiments described here) via MOP should be properly discussed in Discussion and backed-up with appropriate citations (action of DAMGO on midbrain neurons).
13) Line 604. In females?
Minor editing of English language required
Author Response
Response to Reviewer 1 Comments
The study by Boero et al reports on the life threatening effect in stress situation of the FDA approved neurosteroid allopregnanolone ( 3a,5a)3-hydroxy-pregnan-20-one or 3a,5a-THP) recently introduced into clinic for the treatment of post-partum depression. Allopregnanolone (3a,5a-THP) was also tested recently and showed positive effects in rodent models of the cocain-withdrawal syndrome. Effect (passive submersion under the water in vertical position in 20cm –diameter water tank) was detected now in a rat model of forced swim stress (FSS) under 3a,5a-THP (15mg/kg, i.p.). Some, but not all, measured parameters of this – for rats - abnormal behavior were sensitive to the BBB permeable mu-opioid receptor (MOP) antagonist CTAP. Using ELISA and immunoblotting, the authors show increase in b-endorphin and MOP levels after FSS and/or 3a,5a-THP (i.p.) with sex-specific variations (commenting paragraphs in introduction and discussion are identical and should be reformulated). The paper has numerous typos and presents very hard reading. I suggest to present findings which were CTAP-sensitive and those which were not CTAP-dependent separately e.g. in a table instead of numerous bar histograms which are not properly labelled. The finding that i.p. injection of water (vehicle) significantly increases b-endorphin level as well (not only FSS) should be mentioned in the discussion. All known types of stress which will upregulate b-endorphin should be listed. Minimal concentration of 3a,5a-THP combined with minimal concentration of fentanyl (synthetic MOP agonist) to produce LOR (loss of righting reflex) in rat should be determined or a paper cited where it was already done. My specific comments and suggestions for improvements are listed below. In general, I want to emphasize the importance of the finding and high time to see these data published.
We are grateful to the reviewer for the helpful suggestions and we appreciate the time and effort dedicated to providing this valuable feedback on our manuscript. We revised the manuscript to address the suggestions provided by the reviewer. Here is a point-by-point response to the reviewer’s comments.
Major concerns
11) Allopregnanolone is not only an anxiolytic, anticonvulsant and antidepressant drug, as authors mention in their introduction (lines 57-58), but also a general anesthetic (it can produce loss of consciousness or “hypnosis” in animal experiment and spinal areflexy). Supra-synergistic action between etomidate and the neurosteroid 5b-pregnan-3a-ol-20one, which binds to the same site on GABAAR as 3a,5a-THP, was reported in BJP 2014 by Li et al (PMID 25117207). Neurosteroid was injected i.p. in this study at 1mg/kg.
We are unaware of any evidence that allopregnanolone is an anesthetic. Allopregnanolone has anxiolytic and sedative actions, but does not produce hypnosis or anesthesia, even at high doses. We reference the work of others on this point and confirm their observations in this manuscript. We wonder if the reviewer is thinking of alphaxalone, a synthetic analog of allopregnanolone that is a general anesthetic? If we have missed a new finding, please clarify this point for us with a citation.
Therefore, the combined effect of MOP agonist fentanyl and 3a,5a-THP needs to be studied. Best will be an EEG recording to estimate level of anesthesia.
We agree with the reviewer that a pharmacological experiment using sub-threshold doses of fentanyl and allopregnanolone could provide direct evidence for an interaction between allopregnanolone and opiates that would improve our manuscript. However, opioid drugs are strictly regulated in U.S. and we don’ have the authorization to use them. To obtain the authorization, pilot the appropriate sub-threshold doses as well as provide sufficient experimental data would require nearly a year of work. For all those reasons, it’s not possible for us meet the reviewer’s request. We have elaborated more clearly in the discussion that such studies are needed to confirm the interactions suggested by the data in this study.
22) According to the last IUPHAR classification opioid receptors are called MOP, DOP, KOP and NOP (Cox et al, “Challenges for opioid receptor nomenclature: IUPHAR Review 9”, BJP, 2015, 172(2), pp: 317-23). We updated the manuscript with the IUPHAR classification correct name. We updated the manuscript using the IUPHAR classification.
Minor concerns
- Lines 98,99. Last sentence of introduction is awkward and incorrect. Why opioid use disorder? Cocain and alcohol use disorders? Please introduce here your citations 40, 41,42,43,44-46. Same for the last sentence of discussion. We agree with the reviewer’s comment and we modified the last sentence of the Introduction, as well as the Abstract and Conclusion (lines 30-31; 94-96; 739-740).
2) Line 40: Please introduce citations for “Postpartum depression”. We added the definition of PPD in line 41 of the Introduction.
3) Use standard nomenclature for abbreviations, e.g. i.p. instead of IP and a.m. instead of AM. We corrected those abbreviations.
4) “Euthanasie” generally means killing by overdosing general anaesthetics. Please provide more details on your procedures. We added more information in Section 2.3 (line 155)
5) Fig.1 lowest panel. How can behavioral analysis can be done after euthanization? The rats were euthanized by decapitation 20 min after the FSS exposure. We added this detail in Sections 2.3 and 2.5.a (lines 156; 203).
6) Lines 310, 323 defects in punctuation in 3a,5a-THP. We corrected those typing errors.
7) Fig 8 shows no labelling of bars. Figure 8 is now corrected.
8) Line 504. “..CTAP increased the time spent during immobility..” Spent in immobility? Corrected.
9) Space is often missing between drug and dose. Corrected.
10) Line 517: ..in female rats CTAP2mg/kg… reduced the 3a,5a-THP-induced increased in immobility.." We corrected this sentence (lines 508-510).
11) Figures: symbol labelling is misleading. W+VEH is separated by 2 days injections of water (vehicle: VEH) . Authors should also show whether b-endorphin release was similar after repeated injection of VEH or showed habituation. Scheme of W, VEH injection should be added to the Fig.1. We added a group scheme abbreviation in figure 1.
12) Page 549. Behavioral excitation (evident from experiments described here) via MOP should be properly discussed in Discussion and backed-up with appropriate citations (action of DAMGO on midbrain neurons). We respectfully disagree with the reviewer that we found any evidence for behavioral excitation. Our data show an increase in passive beahaviors (submersion and immobility time) and a decrease in active behaviors (swimming).
13) Line 604. In females? We added “in females” in line 596 (section 3.4.6, second paragraph).
Comments on the Quality of English Language
Minor editing of English language required.
We revised the manuscript to correct English language errors.
Reviewer 2 Report
Boero et al. examined the effects of 3α,5α-THP on the behavioral phenotypes elicited by Forced Swim Stress (FSS). Based on the well-established relationship between GABA-A-related neurosteroids and the opioid system, the authors hypothesized that the behavioral outcomes in response to FSS were mediated by the modulation of 3α,5α-THP on opioid signaling. They scored the FSS-induced phenotypes after 3α,5α-THP administration and performed post-mortem analyses of opioid system components, including circulating β-endorphins and hypothalamic Mu-opioid receptors type 1 (MOR1). To establish the role of opioid signaling in the effects of 3α,5α-THP on the behavioral outcomes of FSS, they tested a separate group of animals with a MORs antagonist (CTAP) and found that, overall, the behavioral changes elicited by 3α,5α-THP on FSS involved the contribution of the opioid system through the activation of MOR1.
The paper is well written, the results are well presented (graphs are presented with individual data points), and the statistical analysis is appropriate. Males and females, as well as appropriate dose curves, are included in this investigation. The potential bias/pitfalls of the study are controlled, and the discussion includes the limitations of the study. The state of the art is sufficiently solid to support the hypothesis and take-home message of the study.
I only have a few minor suggestions the authors may wish to address. Please see below.
The abstract needs to be revised.
Please delete the sentence: “we observed submersion in response to 3α,5α-THP (15mg/kg, IP) during forced swim stress (FSS)”….(line 19).
Suggestion: We previously showed 3α,5α-THP down-regulation of HPA axis activity during stress is sex, brain region and stressor dependent. Here, we used FSS to investigate whether and how 3α,5α-THP might affect coping behavioral strategies engaged by the animal to acute inescapable stress. Moreover, given the well-established involvement of opioid system in HPA axis activation and interaction with GABA-A-related neurosteroids, we explored….
Introduction
Please delete the sentence “…and thanks to the contribution to many brilliant scientists (line 37)”
I suggest providing better support for the hypothesis in the introduction.
Suggestion:
Please add this sentence in the line 72. Moreover, it has been previously demonstrated that 3α,5α-THP modulates central opioid expression and produces opioid inhibition over HPA responses (Brunton et al., 2009; DOI: https://doi.org/10.1523/JNEUROSCI.0708-09.2009).
Accordingly, please change this sentence to: “We hypothesized that the behavioral outcomes of FSS might be the result of the fine modulation of 3α,5α-THP on opioid-mediated inhibitory transmission”.
Please remove or change the final sentence of the introduction (line 98). I do not fully agree with this sentence. In my opinion, this study could be clinically relevant in relation to the therapeutic applicability of brexanolone in psychiatric disorders that involve HPA axis activation (as indeed pointed out by the authors in the discussion).
Please change or remove the following sentences: Moreover, at high doses opioids can cause sedation [34]. In addition, both human and animal studies showed the opioid drugs result in an impairment in tasks that require multiple cognitive functions relying on different regional brain circuits [34]. In this study, we showed that blocking Mu-opioid receptor system with CTAP reduced submersion and climbing while increased immobility behavior. This might be the result of a possible recovery of cognitive functions and active coping strategies, through which the animals maximized their effort to survive (lines 685-691).
In my opinion, it is too speculative, and I do not see the connection with cognitive function here, but rather an involvement of the opioid system in shaping appropriate active coping strategies to stress.
Please add to the discussion that FSS is routinely used to screen drugs with antidepressant activity and that only in recent years it has been proposed to test measures of stress-coping strategies, not depression-like phenotypes (Commons et al., 2017 and others...).
Please remove “opioid” from sigma1 opioid receptor effects (line 695). To my knowledge, sigma 1 has been originally proposed to be a subtype of opioid receptors. However, while it binds benzomorphans, the σ receptor is insensitive to naloxone, a universal antagonist of opioid receptors. Sigma-1 is a chaperon that, once activated, it physically interacts with several membrane proteins, ion channels, and G-protein-coupled receptors, including opioid receptors.
Please change this sentence: Please remove "opioid" from sigma-1 opioid receptor effects (line 695). To my knowledge, sigma-1 has been originally proposed to be a subtype of opioid receptors. However, while it binds benzomorphans, the σ receptor is insensitive to naloxone, a universal antagonist of opioid receptors. Sigma-1 is a chaperone activated by multiple systems and, once activated, it physically interacts with several membrane proteins, ion channels, and G-protein-coupled receptors, including opioid receptors.
Author Response
Response to Reviewer 2 Comments
Boero et al. examined the effects of 3α,5α-THP on the behavioral phenotypes elicited by Forced Swim Stress (FSS). Based on the well-established relationship between GABA-A-related neurosteroids and the opioid system, the authors hypothesized that the behavioral outcomes in response to FSS were mediated by the modulation of 3α,5α-THP on opioid signaling. They scored the FSS-induced phenotypes after 3α,5α-THP administration and performed post-mortem analyses of opioid system components, including circulating β-endorphins and hypothalamic Mu-opioid receptors type 1 (MOR1). To establish the role of opioid signaling in the effects of 3α,5α-THP on the behavioral outcomes of FSS, they tested a separate group of animals with a MORs antagonist (CTAP) and found that, overall, the behavioral changes elicited by 3α,5α-THP on FSS involved the contribution of the opioid system through the activation of MOR1.
The paper is well written, the results are well presented (graphs are presented with individual data points), and the statistical analysis is appropriate. Males and females, as well as appropriate dose curves, are included in this investigation. The potential bias/pitfalls of the study are controlled, and the discussion includes the limitations of the study. The state of the art is sufficiently solid to support the hypothesis and take-home message of the study.
We are grateful to the reviewer for the helpful suggestions and we appreciate the time and effort dedicated to providing this valuable feedback on our manuscript. We revised the manuscript to address the suggestions provided by the reviewer. Here is a point-by-point response to the reviewer’s comments.
I only have a few minor suggestions the authors may wish to address. Please see below.
The abstract needs to be revised. We revised the abstract as suggested.
Please delete the sentence: “we observed submersion in response to 3α,5α-THP (15mg/kg, IP) during forced swim stress (FSS)”….(line 19).
Suggestion: We previously showed 3α,5α-THP down-regulation of HPA axis activity during stress is sex, brain region and stressor dependent. Here, we used FSS to investigate whether and how 3α,5α-THP might affect coping behavioral strategies engaged by the animal to acute inescapable stress. Moreover, given the well-established involvement of opioid system in HPA axis activation and interaction with GABA-A-related neurosteroids, we explored….
We accepted the reviewer’s suggestion and addressed this point (line 19-24).
Introduction
Please delete the sentence “…and thanks to the contribution to many brilliant scientists (line 37)”. We deleted this attribution, as suggested.
I suggest providing better support for the hypothesis in the introduction. We improved the Introduction as suggested.
Suggestion:
Please add this sentence in the line 72. Moreover, it has been previously demonstrated that 3α,5α-THP modulates central opioid expression and produces opioid inhibition over HPA responses (Brunton et al., 2009; DOI: https://doi.org/10.1523/JNEUROSCI.0708-09.2009).
Accordingly, please change this sentence to: “We hypothesized that the behavioral outcomes of FSS might be the result of the fine modulation of 3α,5α-THP on opioid-mediated inhibitory transmission”. We accepted the reviewer’s suggestion and we addressed this point (line 78-82).
Please remove or change the final sentence of the introduction (line 98). I do not fully agree with this sentence. In my opinion, this study could be clinically relevant in relation to the therapeutic applicability of brexanolone in psychiatric disorders that involve HPA axis activation (as indeed pointed out by the authors in the discussion). We agree and modified the final sentence as suggested. (lines 94-96).
Please change or remove the following sentences: Moreover, at high doses opioids can cause sedation [34]. In addition, both human and animal studies showed the opioid drugs result in an impairment in tasks that require multiple cognitive functions relying on different regional brain circuits [34]. In this study, we showed that blocking Mu-opioid receptor system with CTAP reduced submersion and climbing while increased immobility behavior. This might be the result of a possible recovery of cognitive functions and active coping strategies, through which the animals maximized their effort to survive (lines 685-691).
In my opinion, it is too speculative, and I do not see the connection with cognitive function here, but rather an involvement of the opioid system in shaping appropriate active coping strategies to stress.
We modified the discussion to remove the suggestion that cognitive functions were involved in the behavior.
Please add to the discussion that FSS is routinely used to screen drugs with antidepressant activity and that only in recent years it has been proposed to test measures of stress-coping strategies, not depression-like phenotypes (Commons et al., 2017 and others...). We added this point in the Discussion, as suggested. (line 654-656).
Please remove “opioid” from sigma1 opioid receptor effects (line 695). To my knowledge, sigma 1 has been originally proposed to be a subtype of opioid receptors. However, while it binds benzomorphans, the σ receptor is insensitive to naloxone, a universal antagonist of opioid receptors. Sigma-1 is a chaperon that, once activated, it physically interacts with several membrane proteins, ion channels, and G-protein-coupled receptors, including opioid receptors. We corrected this designation as suggested (line 685).
Round 2
Reviewer 1 Report
Authors repaired some but not all defects indicated in my previous letter.
1) In a paper I suggested authors should read (Cox et al, Guidelines of IUPHAR on Nomenclature of opioid receptors, PMID 24528283) it is written:
"Beyond μ, δ, κ and NOP receptors, a description of opioid receptor subtypes such as μ1 or μ2 is not recommended unless they are described as putative". Authors still use "MOP1" or "MOR 1" in their manuscript (line25, 33,227,404,410-441,698-705).
2) I accept argumentation that experiments with morphine and allopregnanolone will need long time to obtain animal experiment permission and drugs, this should be done after initial findings are published. Plan to perform such experiments should be added to the conclusions and perspectives, possibility of such drug interaction should be discussed.
3) Allopregnanolone can cause loss of righting reflex which classifies this substance as general anaesthetic (see PMID 33405197,18200328)
4) Figures lettering is too small (needs to be enlarged).
5)Line 22: of the opiate system
6)Use "3α,5α-THP" instead of "ALLO" in Figures 8,9,10
Author Response
Response to Reviewer 1 (round 2)
Authors repaired some but not all defects indicated in my previous letter.
1) In a paper I suggested authors should read (Cox et al, Guidelines of IUPHAR on Nomenclature of opioid receptors, PMID 24528283) it is written:
"Beyond μ, δ, κ and NOP receptors, a description of opioid receptor subtypes such as μ1 or μ2 is not recommended unless they are described as putative". Authors still use "MOP1" or "MOR 1" in their manuscript (line25, 33,227,404,410-441,698-705). We updated the manuscript with the appropriate nomenclature.
2) I accept argumentation that experiments with morphine and allopregnanolone will need long time to obtain animal experiment permission and drugs, this should be done after initial findings are published. Plan to perform such experiments should be added to the conclusions and perspectives, possibility of such drug interaction should be discussed. We agree with the reviewer’s comment, and we added this point in the conclusion (line 739-742).
3) Allopregnanolone can cause loss of righting reflex which classifies this substance as general anaesthetic (see PMID 33405197,18200328). Thank you for these citations. In the literature, the LORR is considered hypnosis, not anesthesia, which requires more profound loss of consciousness (even in the face of surgical intervention). We reason that allopregnanolone does not have hypnotic effects at 15 mg/kg and the work of Reddy’s group (PMID 18200328) that you suggested also confirms this finding in mice, so we have added it to the manuscript on page 2 line 61 and page 3 line 122. PMID 33405197 is another reference for the anticonvulsant actions of allopregnanolone, but intranasal administration is not comparable to IP administration so comparisons to the doses in this paper doesn’t seem appropriate. We hope you accept this explanation. We are certainly open to citing an anesthetic action of allopregnanolone that meets the criterion of other anesthetics, such as alphaxalone or fentanyl, but we are unaware of such data in the literature.
4) Figures lettering is too small (needs to be enlarged). We increased the font in the figures.
5)Line 22: of the opiate system. Corrected.
6)Use "3α,5α-THP" instead of "ALLO" in Figures 8,9,10. We corrected the figures.